# A Comparative Study on Crystallisation for Virgin and Recycled Polyethylene Terephthalate (PET): Multiscale Effects on Physico-Mechanical Properties

**DOI:** 10.3390/polym15234613

**Published:** 2023-12-04

**Authors:** Laurianne Viora, Marie Combeau, Monica Francesca Pucci, Didier Perrin, Pierre-Jacques Liotier, Jean-Luc Bouvard, Christelle Combeaud

**Affiliations:** 1Mines Paris, Centre for Material Forming (CEMEF), PSL University, UMR CNRS 7635, 1 rue Claude Daunesse, CS 10207, 06904 Sophia Antipolis, France; laurianne.viora@minesparis.psl.eu (L.V.); jean-luc.bouvard@minesparis.psl.eu (J.-L.B.); christelle.combeaud@minesparis.psl.eu (C.C.); 2PCH, IMT Mines Ales, 30100 Ales, France; marie.combeau@mines-ales.fr (M.C.); pierre-jacques.liotier@mines-ales.fr (P.-J.L.); 3LMGC, IMT Mines Ales, University of Montpellier, CNRS, 30100 Ales, France; monica.pucci@mines-ales.fr

**Keywords:** recycled PET, crystallinity, thermal and mechanical properties

## Abstract

Poly(Ethylene Terephthalate) (PET) is one of the most used polymers for packaging applications. Modifications induced by service conditions and the means to make this matter circular have to be understood to really close the loop (from bottle to bottle for example). Physico-chemical properties, crystalline organisation, and mechanical behaviour of virgin PET (vPET) are compared with those of recycled PET (rPET). Using different combined experimental methods (Calorimetry, Small Angle X-ray Scattering [SAXS], Atomic Force Microscopy [AFM], Dynamic Mechanical Analysis [DMA], and uniaxial tensile test), it has been proven that even if there is no change in the crystallinity of PET, the crystallisation process shows some differences (size and number of spherulites). The potential impact of these differences on local mechanical characterisation is explored and tends to demonstrate the development of a homogeneous microstructure, leading to well-controlled and relevant local mechanical property characterisation. The main contribution of the present study is a better understanding of crystallisation of PET and recycled PET during forming processes such as thermoforming or Injection Stretch Blow Moulding (ISBM), during which elongation at the point of breaking can depend on the microstructure conditioned by the crystallisation process.

## 1. Introduction

Poly(Ethylene Terephthalate) [PET] is one of the most largely used polymers in food and drink packaging due to its adapted barrier properties, transparency, and ability to be manufactured in films [1]. It is also used in various sport industry applications, like for clothing or equipment (e.g., hang gliders), due to its interesting thermo-mechanical properties [2].

Following massive awareness, recycling of polymers is an important challenge for the plastic industry [3]. PET is the major feed from household plastics (370 Mt of annual global plastic production in 2019) [4]. Consequently, PET is the most common plastic in the natural environment, predominantly shaped as bottles [5]. Moreover, PET is a very inert material that degrades very slowly in nature [6]. Post-consumer plastics must be recovered. Currently, 66% of PET is collected for recycling in Europe [7]. The rest is incinerated or sent to landfill, releasing greenhouse gases. On the contrary, recycling PET would increase circularity and reduce CO_2_ emissions during both production and post-customer processing [8,9].

Two main processes are applied to recycle PET: chemical recycling and mechanical recycling [2,10]. Chemical recycling requires retrieving PET base monomers and purifying them. In a second step, another polymerisation reaction is then needed [11,12,13,14], which enables one to obtain PET of virgin quality that can be used in membrane applications [15]. There are different methods of chemical recycling such as glycolysis, metanalysis, and hydrolysis [2,15,16]. However, chemical recycling is an expensive process that requires the use of catalysts and high temperatures [14,17]. Mechanical recycling is therefore generally used for its ability to be widely used and its low cost. It involves sorting PET, removing contaminants including water by drying, milling, and processing the material to form recycled Poly(Ethylene Terephthalate) [rPET] pellets [18].

Today, rPET is already used for many applications, such as composites [19], but the best-known is textiles (fleeces, microfibers) [5]. There is also a growing interest in the manufacturing of recycled PET bottles [20], since the textile approach is not yet circular. Actually, it remains difficult, for many reasons, to reach 25% of recycled content in all PET bottles by 2025 as imposed by the European Union [21]. A deeper understanding of PET recycling has to be achieved to reach the ultimate goal of circular 100% recycled bottles. Indeed, during the recycling of PET, many parameters must be considered, in particular the presence of contaminants (e.g., other polymers such as poly(propylene), poly(ethylene), poly(vinyl chloride), metals, ink, glues...) and the effect of mechanical recycling (multiple reprocessing cycles), which have an impact on the final properties of rPET [22]. Impurities cause changes in rPET rheology as well as a significant decrease in mechanical properties. Some impurities are not miscible with PET, so they form aggregates that make PET brittle. These impurities can also alter the thermal properties of PET and the transparency of the final product [23,24,25,26,27,28]. In addition, during multiple reprocessing cycles, aging of the PET is observed, leading to a competition between chain scission and chain branching in the polymer, which induces a change in the thermal, mechanical, and rheological properties of rPET [29,30,31,32]. These properties are particularly crucial to optimise the Injection Stretch Blow Moulding process (ISBM) dedicated to bottle production [7,33]. Indeed, in order to obtain recycled transparent bottles, it is necessary to accurately control PET crystallisation during the shaping process through monitoring and adapting the process parameters, inducing suitable service properties [33].

The present study focuses on the relation between the physico-chemical properties of PET and the mechanical properties at different scales to understand the modification of polymer properties after recycling. Three types of PET are used: virgin PET (vPET), recycled PET (100rPET), and a blend of recycled material with virgin (50rPET) at 50/50%. This comparison allows us to better understand the ability of recycled PET to develop a specific microstructure when being transformed through forming processes such as film stretching, thermoforming, or even the ISBM process. All these processes require high molecular weights, supposing long chains which are highly entangled, thus promoting efficient texturation and Strain-Induced Crystallisation (SIC). Intrinsic parameters like the molecular weight or the molecular weight distribution are of prime interest and are explored to understand the microstructural mechanisms involved in crystallisation processes.

Dynamic crystallisation under stretching is not explored. Nevertheless, in static conditions, it is also possible to deeply explore chain mobility, their ability to develop organised structures that will be relevant for their final application. An analysis of thermal properties using Differential Scanning Calorimetry (DSC) coupled to microscopic observations is then proposed. Spherulitic structures developed are described thanks to crystalline lamellae thickness measurements relying either on calorimetry (DSC) or on Small-Angle X-ray Scattering (SAXS) analysis. To complete these morphological descriptions, local mechanical tests were performed by Atomic Force Microscopy (AFM) where mechanical signatures of crystalline lamellae could be analysed. More classical mechanical analyses like Dynamic Mechanical Analysis (DMA) and uniaxial tensile tests are used to corroborate the previous experiments at the macroscopic scale. To our knowledge, no contribution to the literature has yet proposed such a complete approach in the case of recycled PET, drawing the link between microstructural organisation and mechanical properties.

## 2. Materials and Methods

### 2.1. Materials and Processing Protocols

Virgin PET (vPET) and recycled PET (100% recycled PET, referred as 100rPET later in the text) pellets were provided by Indorama Ventures^®^ (Bangkok, Thailand) and LPR^®^, respectively. The two polymer resins chosen are commonly used in injection moulding processes. Preforms can then be injected and, for a second time, subjected to ISBM. Blends of vPET and 100rPET were achieved by injection moulding [26,34,35] according to industrial procedures, with an equivalent weight of vPET and 100 rPET, leading to a blend of 50% virgin and 50% recycled PET referred as “50rPET”. A typical geometry used for ISBM was chosen. Sidel Group supplied preforms with the following characteristics: 18.5 g weight, 2.55 mm thickness, 95.2 mm length, and 2.0 mm major diameter. The polymer pellets have been dried under vacuum before injection to avoid hydrolysis during the processing step. The mixture of polymer blends involved in the injection process is representative of the industrial process. It is obvious that this kind of mixing is less efficient compared to the extrusion process [36]. However, the experimentations showed that all the materials had a chemical composition sufficiently near that required to obtain a homogeneous mixture. The same injection process parameters have been used for each of the three materials. The goal was to rapidly cool down the melt to quench the microstructure in an amorphous state. As a matter of fact, these injection protocols are common in industrial contexts where SIC has to take place during the blowing process in a second step.

Moreover, tensile test specimens have also been manufactured thanks to a mini-press, using PET dried for 6 h at 140 °C. Indeed, to process the PET pellets, cylinders were heated to 270 °C and 3 bar of pressure was applied for 5 s to fill the moulds. The crystallinity rates in those samples were increased by injecting in a hot mould. To achieve transparent samples, it is common to inject in moulds at 20 or 60 °C, below the PET glass transition temperature (T_g_ = 67–140 °C [10]). In the present study, the mould was set to 140 °C, allowing crystallisation of the polymer from the melt state. This procedure results in opaque samples, proving their increased crystallinity rate.

This work focused on crystallisation and associated morphologies. To study this phenomenon, amorphous preforms of vPET, 50rPET, and 100rPET were placed in an oven at 140 °C for 24 h in order to develop cold crystallisation. Then, parallelepipedal specimens were machined from the body of the preforms. Such samples have been used for different analysis methods such as DSC, SAXS, and DMA as described below. As a result, except in the case of PET pellets, only semi-crystalline preforms will be analysed and discussed in the following steps.

### 2.2. Melt Flow Index (MFI)

The Melt Flow Index or Rate (MFI or MFR) is an industrial parameter related to the viscosity of a molten polymer. MFI allows one to define the grade of a polymer (injection- or extrusion-grade). Using pellets, the MFI of vPET and rPET were measured thanks to an INSTRON CEAST MF20 melt flow tester as defined in the standard test method ISO 1133. The measurements were performed at a temperature of 270 °C. Thirteen recordings were taken to determine the average MFI of each PET.

### 2.3. Size Exclusion Chromatography (SEC)

Size Exclusion Chromatography (SEC) is an accurate method to estimate the molecular weight (in number average, Mn, or in weight average, Mw) of polymers. The Polydispersity Index (PDI) is defined as the ratio of Mw and Mn. If the value is close to 1, the molecular weight distribution is low. The higher the value is, the larger the Gaussian distribution is (up to 3).

First, the pellets were dissolved in Hexafluropropan-2-ol (HFIP) to obtain a concentration between 3 and 7 mg/mL. The solution was then filtrated (on 0.45 µm filters) and analysed through the SEC device (System Waters APC Acquity, Waters corporation, Milford, MA, USA) with four columns at 35 °C. Molecular weights were determined by light diffusion with a MALS WYATT (microDawn) device (Waters corporation, Milford, MA, USA). Two measurements were carried out on both virgin and recycled pellets. All those experiments were carried out by the INSA Valor company.

### 2.4. Polarised Optical Microscopy

The observations were conducted on a Hot Stage Leica DM4500P microscope (Leica Microsystems, Wetzlar, Germany) with a magnification of ×200. They were performed using crossed polarisers with a gypsum plate. To analyse the development of spherulites over cold crystallisation phenomenon, a completely amorphous sample was prepared. Small parts of a pellet were collected with a razor blade. Then, the material was melted between two glass slides at 300 °C for five minutes. The sandwich system was cooled down in water at ambient temperature to quench the microstructure in its amorphous state. The obtained samples were reheated at 20 °C/min until reaching an isothermal step at a temperature of 120 °C where cold crystallisation may occur.

### 2.5. Differential Scanning Calorimetry (DSC)

The thermal properties of each material were analysed using a PerkinElmer Pyris diamond DSC (Waltham, MA, USA) under a nitrogen flow. Samples of 4 ± 0.5 mg were collected in the central zone of semi-crystalline preforms and were encapsulated in aluminium pans.

Figure 1 displays the heating and cooling steps applied for DSC analysis. 

A heating ramp of 10 °C/min up to a temperature of 290 °C was first applied. After a holding step at 290 °C for one minute, a cooling step with a ramp of 10 °C/min to reach 30 °C was performed. Samples were maintained at this temperature for one minute. Finally, a second similar heating ramp was conducted. The glass transition temperature, T_g_, was determined drawing two tangents at temperatures below and above the main transition. The cold crystallisation, hot crystallisation, and melting temperatures, T_cc_, T_ch_, and T_m_, were obtained at the maximum of exothermic (from the cold and melting state) and endothermic enthalpy peaks, respectively. Associated enthalpies, ΔH_cc_, ΔH_ch,_ and ΔH_m_, were determined from the heating scans. T_mi_ are given as T_m_ for the crystalline population i. The crystallinity ratio from heating thermograms, X_c,H_, was calculated according to Equation (1):(1)$$X_{c,H}\;(\%)=\frac{\Delta H_{m}-\Delta H_{cc}}{\Delta H_{0}}\times 100$$
where ΔH_m_ represents the melting enthalpy (J/g), ΔH_cc_ the cold crystallisation enthalpy (J/g), and ΔH_0_ = 140 J/g the melting/cooling enthalpy of 100% crystalline PET [26,31,33,37,38].

The crystallinity ratio from cooling thermograms, X_c,C_, was calculated according to Equation (2):(2)$$X_{c,C}\;(\%)=\frac{-\Delta H_{ch}}{\Delta H_{0}}\times 100$$
where ΔH_ch_ represents the cold crystallisation enthalpy (J/g) and ΔH_0_ = 140 J/g the melting/cooling enthalpy of 100% crystalline PET [26,31,33,37,38].

Three tests have been conducted for each material (vPET, 50rPET, and 100rPET).

### 2.6. Small-Angle X-ray Scattering (SAXS)

The SAXS analysis was performed with an Empyrean diffractometer with CuKα radiation, presenting a wavelength of 1.54 Å at room temperature and under 45 kV and 30 mA. Angular scans were conducted at very small angles, between 0.1° and 2° (2*θ*), allowing the analysis of all planes of diffusion between crystalline lamellas. The *θ* angle was corrected by the Lorentz factor because of the phenomenon of diffusion. The Lorentz correction consists of the multiplication of the scattering intensity by a factor proportional to the sine of the diffraction angle. The invariant *Q* is defined as follows (Equation (3)):(3)$$Q=q^{2}\times {\;}_{measured}\left(q\right)={\left(\frac{4\pi sin\left(\theta \right)}{\lambda }\right)}^{2}\times I_{measured}\left(q\right)\;\;$$
where *q* is the scattering factor and *I_measured_* is the measured scattered intensity before correction [37,38].

The long period *L* was calculated using the diffraction Bragg’s law (Equation (4)).
(4)$$L=\frac{\lambda }{2\;sin\left(\frac{\theta_{corr}}{2}\right)}$$
where *λ* is the X-ray wavelength and *θ_corr_* is the scattered angle after correction with the Lorentz factor [39,40]. As for DSC experiments, the specimens were extracted in the same central zone of crystallised preforms. Parallelepipedal specimens have been extracted from the body of the preforms and one unique SAXS analysis has been performed for each of the three materials.

### 2.7. Atomic Force Microscopy (AFM)

Microstructural characterisations were carried out via an AFM on preforms of 100% virgin PET (vPET), 100% recycled PET (100rPET), and the blend (50rPET). Prior to AFM analyses, samples were prepared using a Leica EM UC7 ultra-cryomicrotome, allowing us to obtain very flat and smooth surfaces. The collecting took place in the same central zone of the preforms as the one used for DSC experiments. Then, the MFP-3D Infinity AFM from Asylum Research (Oxford Instruments, Abingdon, UK) was used in a bimodal tapping mode, also known as Amplitude Modulation–Frequency Modulation (AM-FM) mode [41,42]. This mode consists of a nano-mechanical characterisation. It allows one to obtain the sample topography (height) and the phase contrast as in a tapping mode. In addition, it allows one to obtain the indentation and the local contact modulus. A silicon probe (AC160TS-R3) with a resonant frequency of about 300 kHz and a spring constant of about 25 N/m was used. Before testing the samples, the tip was calibrated on a poly(styrene) film with a known modulus of 2.7 GPa (Bruker Corporation, Billerica, MA, USA) used as reference material. After that, AFM tests with the calibrated tip were carried out on cryomicrotomed surfaces using a scan rate of 1 Hz. Three images of 3 × 3 µm^2^ with a resolution of 256 × 256 pixels were recorded for each sample.

### 2.8. Dynamic Mechanical Analysis (DMA)

Dynamic mechanical analyses were conducted using a Mettler-Toledo^®^ DMA 1. A tensile mode was applied on samples extracted from preforms with a parallelepiped shape of 5.0 × 2.8 × 1.6 mm^3^. The core of the preforms was collected thanks to machining, removing the surrounding parts. A second step which consisted of polishing the two surfaces (inside and outside) was added to obtain a parallelepiped shape, representative of the core of the preform. Tests were performed at a frequency of 1 Hz, with a preload of 1 N and a dynamic deformation of 0.04%. Temperature scans between 25 °C and 140 °C were performed at a heating rate of 2 °C/min. Viscoelastic properties such as the storage modulus (*E*′), the loss modulus (*E*″), and the associated Tan δ = *E*″/*E*′ were measured. The alpha transition Tα was identified at 1 Hz at the maximum of Tan δ peak. Three tests were carried out for each material (vPET, 50rPET and 100rPET). An average curve was then obtained, and the standard deviation was calculated and represented on the curves.

### 2.9. Uni-Axial Tensile Test

Monotonic uniaxial tensile tests until failure were performed using a Zwick Roell Z010 testing machine. The tests were conducted on previously manufactured ISO 527-1BA specimens. Experiments were performed at room temperature with a 3 N preload, a crosshead speed of 5 mm/min, and an initial gauge length of 40 mm. The estimated initial strain rate was of 2.10^−3^ s^−1^. Young’s modulus, *E*, was measured with an extensometer at a strain ranging from 0.05% to 0.25% according to the standard test method ISO 527. The maximum stress, *σ_max_*, and the strain at the maximum stress, *ε*(*σ_max_*), were estimated. Three samples of each material were tested.

## 3. Results and Discussion

### 3.1. Molecular Weight and Rheological Properties

Table 1 summarises the molecular weights, Mn and Mw, as well as the PDI measured for the two polymers by SEC as a comparison of molecular organisation between virgin and recycled PET pellets.

According to data from Table 1, the mass average, Mw, is significantly lower for recycled PET compared to virgin PET, while the numbered average molecular weight, Mn, remains unchanged. These first results show that polymer chains are shorter in the case of the recycled PET. Moreover, the recycled PET shows a lower PDI (Table 1), meaning a lower molecular weight distribution for recycled PET, which could allow for better processing control [10,43]. To complete this analysis, MFI measurements were conducted to confirm or not the hypothesis of shorter polymer chains for the 100rPET. The results of these measurements are summarised in the Table 2.

Such an order of magnitude for MFI values provided in Table 2 is expected due to the low viscosity of PET. As a matter of fact, low viscosities are required to inject preforms that will be blown during the ISBM process. These PET grades are thus commonly used for injection processes, allowing for efficient filling of the preform mould.

The recycled PET presents a significantly higher MFI than the virgin one, which proves lower viscosities. These lower rheological properties are clearly in adequation with the lower molecular weight observed for the recycled PET. Thus, recycling process possibly coupled to aging under service conditions might induce this decrease in chain lengths.

### 3.2. Crystalline Morphologies Observation and Thermal Behaviour Analysis

These very first analyses permit us to roughly describe the molecular organisation of each material. The possible shorter lengths and the presence of contaminants inside the recycled PET are of prime interest to understand the ability of these materials to crystallise in quiescent conditions. To focus on the understanding of cold crystallisation processes, the spherulitic microstructures induced in controlled conditions have been observed thanks to optical microscopy. During an isothermal step imposed at a temperature of 120 °C, the formation and growth of spherulites can be followed. Figure 2a–c shows optical pictures captured at the end of the imposed isothermal step of 120 °C: crystallised samples of vPET, 50rPET, and 100rPET are, respectively, presented below.

Spherulitic organisations are completely different between virgin and recycled PET, as described by Kim and al. [44]. Morphologies found in the blend 50rPET are equivalent to the pure recycled material. In the last case, more numerous spherulites are present, testifying to a higher efficiency in the germination process. Contaminants that are present in the recycled polymer associated with shorter chains lengths do play an important role as local nucleating agents. As a result, the growth is limited by the presence of more numerous activated germination sites. The final diameters of the spherulites are in the order of magnitude of 4 µm or even less, whereas neat PET presents larger entities, with diameters reaching values of around 10 µm.

Moreover, characteristic times of appearance of spherulites for each sample have been estimated, following the thermal procedure described in Section 2.4. The time departure of cold crystallisation at 120 °C thus occurs after different durations during the isothermal step: 380, 190, and 225 s for, respectively, vPET, 50rPET, and 100rPET. These measurements confirm that the recycled PET has a higher ability to crystallise in quiescent conditions. This is a crucial point that must be considered for processability towards the ISBM process. As a matter of fact, for the initially amorphous PET, the forming range above the α-transition can be reduced on the upper limit because of cold crystallisation, which does reduce stretchability and transparency of the polymer. To complete these observations, thermal analyses performed by DSC have been conducted.

Thermograms obtained by DSC for vPET, 50rPET, and 100rPET are presented Figure 3 and Figure 4. As a reminder, semi-crystalline preforms, reheated in an oven to activate cold crystallisation, are characterised. It is also the case of the samples tested in the mechanical section. 

Figure 3 displays during the first heating stage an endothermal peak for all materials. Two melting temperatures are also observed for all materials. The first melting enthalpy is observed at approximately 130 °C (population 0) while the second one is observed at around 245 °C (population 2). Crystallites usually melt at higher temperatures, at least a temperature of 250 °C, without any intermediate melting phenomenon. A less stable microstructural organisation could be present; the existence of a transcrystalline phase has already been reported [45,46]. Moreover, the crystallinity ratio X_c,H_ has been estimated from both the observed melting enthalpies. The values obtained are in the order of magnitude of 33, 35, and 37% for vPET, 50rPET, and 100rPET, respectively, in the range of melting temperatures for virgin and recycled PET [42]. The recycled PET appears clearly more crystalline than the virgin one, while the blended one remains in between.

These observations must be confronted with the second heating step, where the thermo-mechanical history of each of the material has been erased. Concerning Figure 4, several observations can be mentioned. Firstly, the glass transition temperature is changing slightly. The 100rPET polymer presents a T_g_ of 80 °C, which is one degree higher than the virgin one. Secondly, the less stable phase attributed to possible transcrystalline organisation is not detected at all. This observation highlights the fact that thermo-mechanical history induced especially by the injection process may have played a role in pre-organising a local microstructure. The differences observed are thus not intrinsic to the material itself. Thirdly, the cold crystallisation is only observable for the vPET polymer at a temperature of 130 °C. This is a crucial point which proves that amorphising PET requires more severe cooling conditions for recycled PET than for virgin PET. Thus, the microstructure of 100rPET is difficult to quench in an amorphous state, testifying to its ability of crystallising efficiently. Previous studies have reported that faster crystallisation of recycled PET can be attributed to shorter chains, acting as nuclei promoting the growth of more crystalline areas for recycled PET compared to the neat one [37,45,47]. Contaminants present in recycled PET may also play this role. Finally, a split of the melting peak can be seen for all rPET (50rPET and 100rPET): population 1 around T_m1_ and population 2 around T_m2_. Population 1 corresponds to the melting of primary crystals and population 2 results in the reorganisation of crystals supposed to be less perfect [48].

Figure 5 shows a single crystallisation peak for all materials during cooling stage. Crystallisation ratios extracted from the cooling thermograms (X_c,C_ ) and the heating thermograms (X_c,H_ ) match.

### 3.3. Crystalline Morphologies Description: Lamella Thickness Measurements

To go further in the understanding of the morphological crystalline organisation, DSC and SAXS analyses were carried out to observe trends on crystalline lamella thicknesses. Double melting peaks as observed previously can suggest different crystalline lamella populations whose thicknesses can differ from one material to another. Crystalline lamellar thicknesses, corresponding to each endothermal peak, are then investigated using the Gibbs–Thomson relation (Equation (5)):(5)$$l_{c}\left(T_{m}\right)={\left[\left(1-\frac{T_{m}}{T_{m}^{0}}\right)\frac{\Delta h_{mV}}{2*\sigma_{e}}\right]}^{-1}$$
where *T_m_^0^* is the equilibrium melting temperature of an infinite crystal (564 K); σ_e_ is the surface free energy of the basal plane where the chains fold (0.106 J.m^−2^); and *Δh_mV_* is the melting enthalpy per volume unit (2.1 × 10^8^ J.m^−^^3^) [48,49].

Over the first heating step, two populations of crystalline lamellas, population 0 and population 2, have been denoted, as described in Figure 3. Figure 6 show an estimation of crystalline lamella thicknesses calculated for the first heating stage for the three materials to have information about objects studied by AFM and during mechanical tests.

Crystalline lamella thicknesses obtained for population 0 and population 2 are in the order of magnitude of 3.5 to 12 nm, for semi-crystalline PET crystallised in quiescent conditions [49,50]. As expected, 50rPET and 100rPET present slightly higher lamellar thicknesses compared to the virgin sample. Those values are consistent with literature where mechanically recycled PET has higher melting temperatures, associated with more stable crystalline structures [44]. As lamellar thickness is linked to the melting temperature according to the Gibbs–Thomson equation (5), this increase in temperature is associated with thicker crystalline lamellas. The significance of this difference can be questioned since the dispersion is of the order of the differences. More precise characterisation will be conducted in further studies to verify this statement.

These analyses were completed thanks to X-ray scattering, with a SAXS protocol where long periods, L, were measured. In a second time, an order of magnitude of the crystalline lamella thicknesses can then be deduced from long period measurements according to the degree of crystallinity previously analysed by DSC during the first heating stage. A crystallinity ratio from wide-angle X-ray scattering would have been more suitable to compare the absolute value. This approach could be developed in further investigations. Only trends allowing a comparison of the three materials are proposed. Figure 7 presents SAXS diffractograms obtained for the three materials, involving the Lorentz correction presented in the Section 2.6.

The diffractograms obtained including the Lorentz correction clearly show that a periodicity exists inside the three polymers at the scale the of arranged lamellas. This periodicity results in a gaussian distribution of the long period, *L*. The maximum of the peak is, for the three cases, centred on values in the order of magnitude of 60 nm which are common for polymeric systems. From those analyses, and thanks to crystallinity ratios *X_cH_* measured by DSC with the two melting peaks, it is possible to determine an order of magnitude of crystalline lamella thicknesses and associated trend by the multiplication of the long period by the *X_cH_*, as presented on Figure 8.

Thanks to this approach, crystalline lamella thicknesses can be estimated and compared, as the protocols used were rigorously the same from one polymer to another. In Figure 8, the order of magnitude of the lamellar thickness is of 25 nm. Recycled PET (100rPET) could appear as presenting slightly thicker crystalline lamellas. The case of 50rPET seems to stay in between the totally virgin and recycled PET. However, those differences cannot be considered as accurately significant since dispersions are larger than differences between characterised values. To conclude on this part, the crystallisation process is clearly more efficient for recycled PET, with melting peaks observed at higher temperatures. Moreover, non-significant differences can be highlighted regarding crystalline morphological descriptions of virgin and recycled PET which could present slightly thicker crystalline lamellas, but it has not been confirmed in the present study. In the following part, and to go further in an understanding of local microstructural organisation, efforts were made to also describe local mechanical behaviours at the scale of crystalline lamellas.

### 3.4. Mechanical Behaviour at the Mesoscopic Scale

A local mechanical approach is presented thanks to dynamic mechanical characterisation performed by AFM. Mechanical investigations are then conducted at the meso-scale of crystalline lamellas. The analysed surface is of 3 × 3 µm^2^ which corresponds—considering spherulitic diameters of approximately 10 µm or 4 µm for, respectively, vPET and rPET—to a scanning surface of 11 and 72% of a whole spherulitic entity. In other words, this tapping approach consists of analysing a representative part of a spherulite, whose crystalline lamellas are in the order of magnitude of 10 to 60 nm, involving both amorphous and crystalline domains.

Figure 9, Figure 10 and Figure 11 show images obtained with the AM-FM mode described in Section 2.7. Pictures are organised as follows: topography (a), indentation (b), and the contact modulus (c). A histogram with Gaussian fit showing the disparity of elastic moduli has also been plotted (d) for each material. Topography images (Figure 9a, Figure 10a and Figure 11a) show the microstructure of each material and the associated roughness (height). It is possible to observe that, as required for mechanical tests, each surface prepared by the ultra-cryomicrotome is very smooth (height is in the order of 10 nm) allowing one to perform nano-mechanical analysis without artefacts due to the surface roughness. Images of indentation (Figure 9b, Figure 10b and Figure 11b) show that tip penetration is, in all cases, between 300 pm and 2 nm according to the Hertz hypothesis in an intermittent bimodal test. Images of contact modulus (Figure 9c, Figure 10c and Figure 11c) show the local variation of each material stiffness. Distribution of elastic moduli is represented on a histogram and the associated Gaussian fit for each case is provided (Figure 9d, Figure 10d and Figure 11d). The Gaussian fit was successfully applied to each histogram with an R-squared value greater than 0.9. Contours observed in Figure 9 and Figure 11 for vPET and 100rPET, respectively, could represent spherulites edges with interspherulitic regions. This hypothesis would have to be verified in further works.

The elastic moduli histograms display a distribution of mechanical properties which appears symmetrical both in the case of vPET and 100rPET. This symmetrical distribution could be the reflection of both soft and stiff contributions that could be attributed to either amorphous or crystalline domains. The averaged value on these distributions is the same (2.13 GPa) for vPET and 100rPET, even if the fully recycled PET exhibits a higher crystallinity ratio (X_c_ = 36.6%) than the virgin one (X_c_ = 32.9%) and does present possibly thicker crystalline lamellas. No mechanical signature of the presence of more crystalline regions is detectable. The fact that crystalline lamellas would be thicker in the case of recycled PET is not obvious regarding these measurements, too. Indeed, local mechanical signatures appear equivalent, even in terms of distribution. It is of prime interest to note that, finally, rPET is able to develop a well-controlled microstructure as homogeneous as virgin PET regarding the distribution of crystalline morphologies and their local mechanical characteristics. The presence of contaminants, associated to shorter chains, does not seem to affect at all the mechanical properties of the recycled polymer. Moreover, the blend of vPET and 100rPET materials (50rPET) exhibits a singular behaviour, as a less symmetrical and larger distribution of elastic modulus is observed, with an averaged value of 3.03 GPa. The fact that two PET types are blended may have induced possible local non-homogeneous domains. Nevertheless, the mechanical behaviour of the blend remains close to both virgin and recycled PET.

### 3.5. Mechanical Behaviour at the Macroscopic Scale

Characterisations have so far focused on microscopic and mesoscopic scales. The mechanical behaviour of vPET, 50rPET, and 100rPET were studied through both DMA and tensile tests experiments to link properties at the micro- and meso-scale to macroscopic mechanical properties more relevant to the process scale.

Figure 12 and Figure 13 represent, respectively, the evolution of storage modulus and Tan δ as a function of temperature. Values of peaks of Tan δ are given with a precision of 1 °C.

In Figure 12 and Figure 13, the values of storage modulus and Tan δ at room temperature are very similar for all materials, i.e., no significant differences in the mechanical behaviour have been pointed out. More dispersion in the measurements is observed in the case of 50rPET, which is a blend of vPET and 100rPET. Such an observation agrees with AFM measurements regarding the elastic moduli distribution; the peak of elastic modulus is larger for 50rPET than for neat materials. It confirms the fact that the blend presents less homogeneous mechanical properties than the neat or totally recycled PET. Furthermore, in Figure 12, the inflexion point between the glassy and the rubbery-like behaviours appears for higher temperatures in the case of 100rPET, but the shift is of little significance (differences of the order of magnitude of the measurement precision). This is in accordance with the maximum of Tan δ peak (Figure 13) which is shifted to higher temperatures, but it cannot be reliably proven with the present measurements. The blend, 50rPET, still appears in between the two-extremal cases.

According to Figure 13, the area under Tan δ peak is also lower in the case of rPET which proves that the amorphous phase of 100rPET presents less molecular mobility. The presence of more crystalline regions, as suggested by DSC analysis, is constraining the amorphous phase. Moreover, the rubbery-like plateau (Figure 12) appears as slightly higher in the case of 100rPET, which is the most crystalline material, followed by 50rPET and then vPET. Nevertheless, at room temperature, it is obvious that all the PET varieties explored in this study exhibit similar mechanical behaviours.

To complete the viscoelastic property measurements, uniaxial tensile tests have also been conducted on samples manufactured with vPET, 50rPET, and 100rPET. Figure 14 shows a comparison of the stress–strain curves for each material. A representative curve for each sample was chosen to make the comparison after checking the repeatability of those tests with three samples.

This last approach tends to highlight the fact that no significant differences in the mechanical behaviour at the macroscopic scale can be seen between the three polymers. The same trends are observed, for all curves, with a strain at the maximum stress at 5% of deformation [51]. As described in Figure 15, samples exhibit similar moduli for each material (about 2800 MPa) [52]. 

Table 3 summarises values of the modulus obtained with AFM, DMA, and uniaxial tensile tests.

## 4. Conclusions

The purpose of this comparative study between semi-crystalline virgin and recycled PET was to understand the link between physico-chemical properties, morphological description of crystalline entities, and mechanical behaviour.

It is demonstrated that recycled PET (100rPET) presents lower molecular weight compared to virgin PET (vPET), while the Polydispersity Index remains equivalent. The final polycondensation step is efficient for the studied materials. This decrease in chain length is also confirmed by MFI value.

It appears that this aspect coupled with contamination has an impact on crystallisation processes. Indeed, rPET crystallises faster and more easily than vPET. Moreover, its quenching ability decreases. The blend has a similar behaviour compared to the neat recycled PET.

The impact of these differences of crystalline morphologies on local mechanical behaviour was then characterised by AFM. Modulus analysis did not show significant differences between vPET and rPET. The distribution of mechanical responses appears very similar and well-controlled in the rPET bulk. The same trends are obtained with studies at a larger scale. At room temperature, the mechanical responses obtained with DMA and uniaxial tensile tests are equivalent for recycled and virgin materials. Difference of molecular weight and crystallinity does not seem to have any impact on the macroscopic mechanical behaviour of cold crystallised PET.

## Figures and Tables

**Figure 1 polymers-15-04613-f001:**
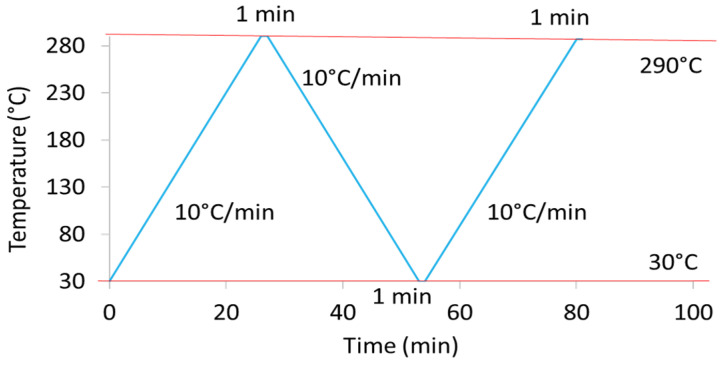
Procedure of heating and cooling steps applied during Differential Scanning Calorimetry (DSC) experiments.

**Figure 2 polymers-15-04613-f002:**
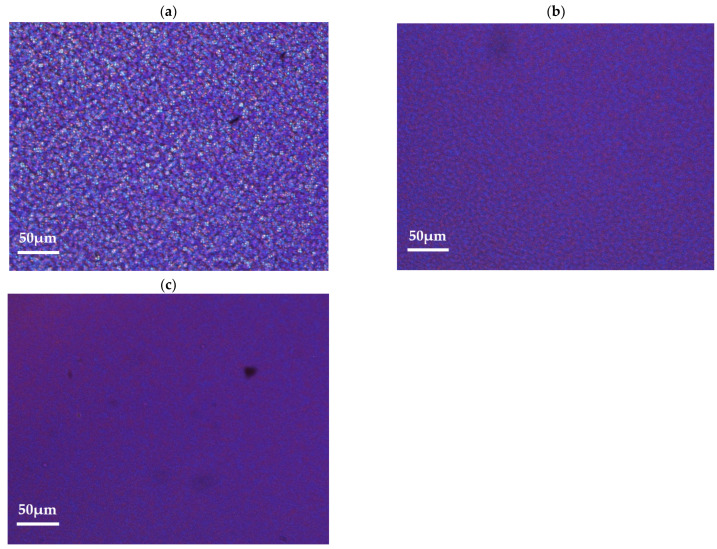
Optical observations of spherulites developed in quiescent conditions, at a temperature of 120 °C, for vPET (**a**), 50rPET (**b**), and 100rPET (**c**).

**Figure 3 polymers-15-04613-f003:**
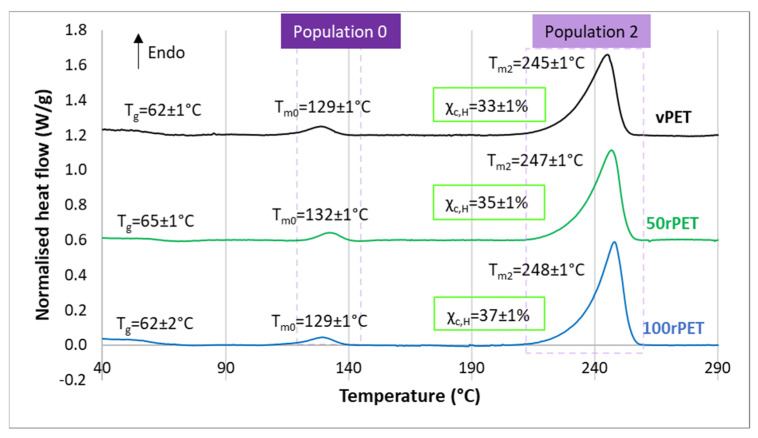
DSC curves of semi-crystalline preforms for vPET, 50rPET, and 100rPET, heated from 30 °C to 290 °C at a heating rate of 10 °C/min (first heating step). Thermograms are vertically shifted.

**Figure 4 polymers-15-04613-f004:**
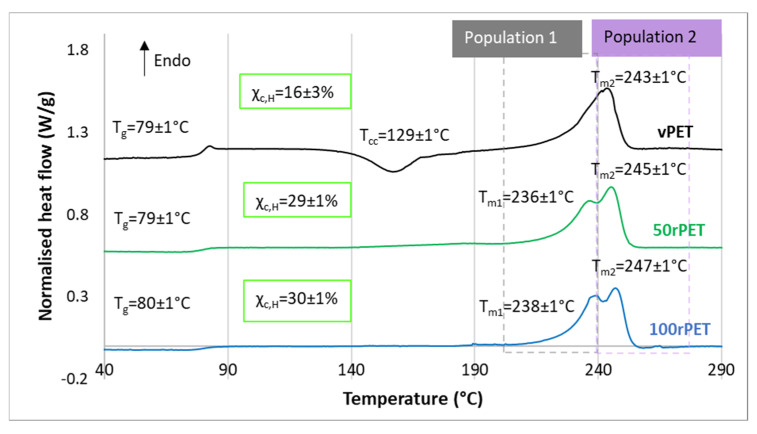
DSC curves of semi-crystalline preforms for vPET, 50rPET, and 100rPET, heated from 30 °C to 290 °C at a heating rate of 10 °C/min for the second heating step after cooling from 290 °C to 30 °C at a cooling rate of −10 °C/min. Thermograms are vertically shifted.

**Figure 5 polymers-15-04613-f005:**
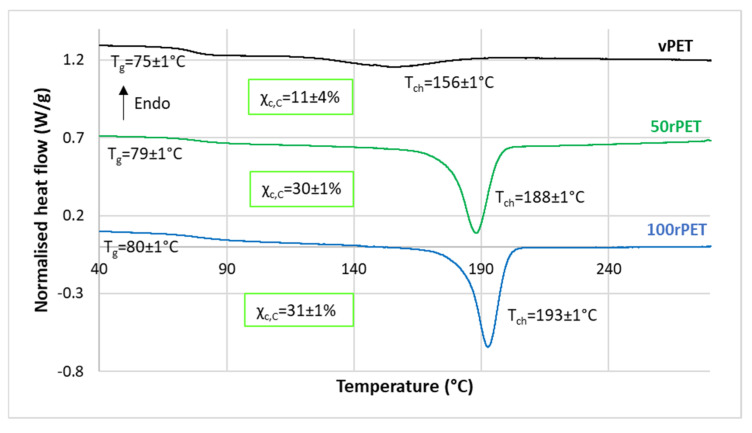
DSC curves of semi-crystalline preforms for vPET, 50rPET, and 100rPET cooled from 290 °C to 30 °C at a cooling rate of −10 °C/min. Thermograms are vertically shifted.

**Figure 6 polymers-15-04613-f006:**
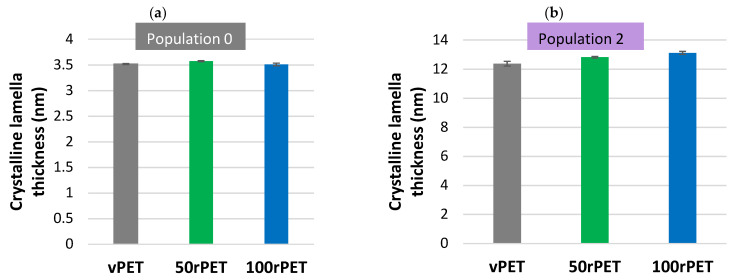
Crystalline lamella thicknesses, determined by DSC measurement, during the first heating step at 10 °C/min, for the lamella population 1 (**a**) and population 2 (**b**).

**Figure 7 polymers-15-04613-f007:**
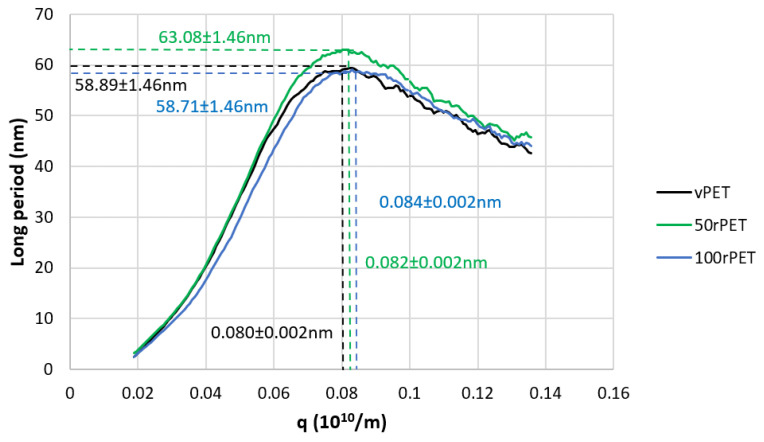
Diffractograms obtained by Small-Angle X-ray Scattering (SAXS) with the Lorentz correction for vPET, 50rPET, and 100rPET associated to semi-crystalline preforms at room temperature.

**Figure 8 polymers-15-04613-f008:**
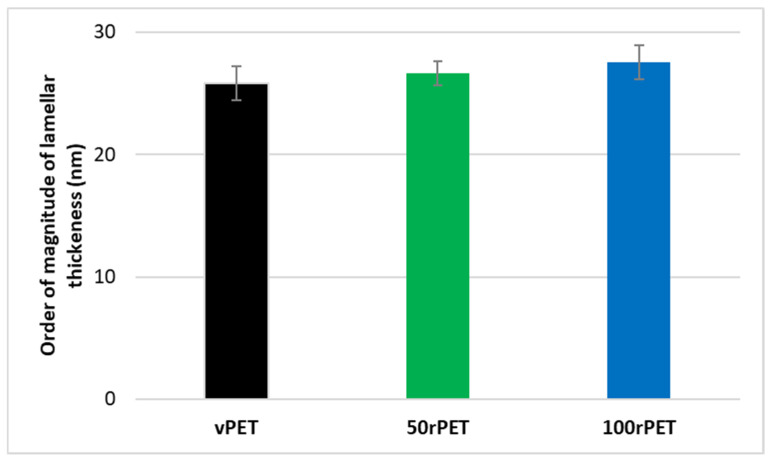
Crystalline lamellar thicknesses for vPET, 50rPET, and 100rPET in semi-crystalline preforms associated to long period, determined by SAXS analysis, multiplied with crystallinity ratios X_cH_ measured by DSC on the first heating thermograms.

**Figure 9 polymers-15-04613-f009:**
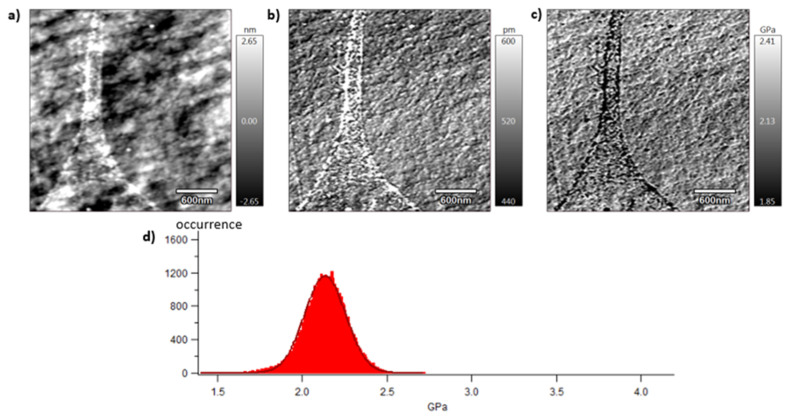
Atomic Force Microscopy (AFM) images of vPET with topography (**a**), indentation (**b**), contact modulus (**c**), and histogram of elastic moduli (**d**) at room temperature.

**Figure 10 polymers-15-04613-f010:**
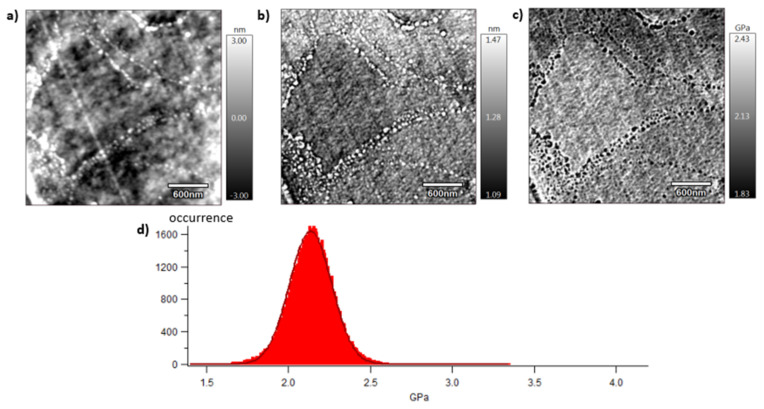
AFM images of 100rPET with topography (**a**), indentation (**b**), contact modulus (**c**), and histogram of elastic moduli (**d**) at room temperature.

**Figure 11 polymers-15-04613-f011:**
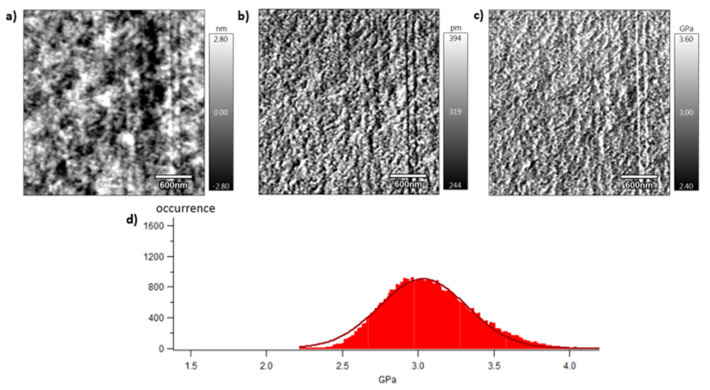
AFM images of 50rPET with topography (**a**), indentation (**b**), contact modulus (**c**), and histogram of elastic moduli (**d**) at room temperature.

**Figure 12 polymers-15-04613-f012:**
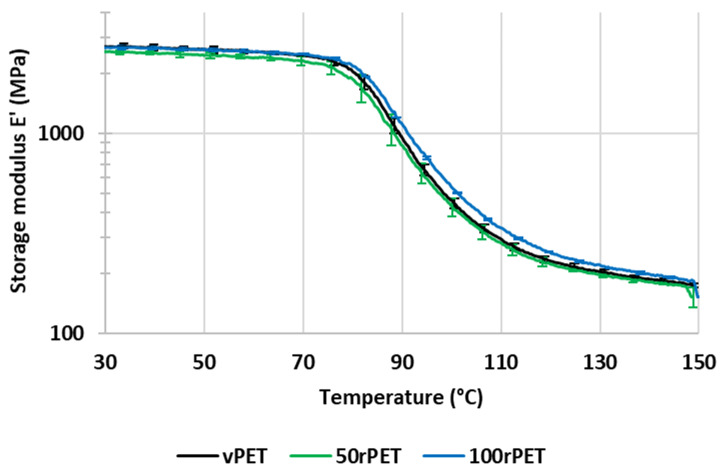
Storage modulus (E′) as a function of temperature for vPET, 50rPET, and 100rPET measured by DMA, in tensile mode, at a frequency of 1 Hz and a heating rate of 2 °C/min.

**Figure 13 polymers-15-04613-f013:**
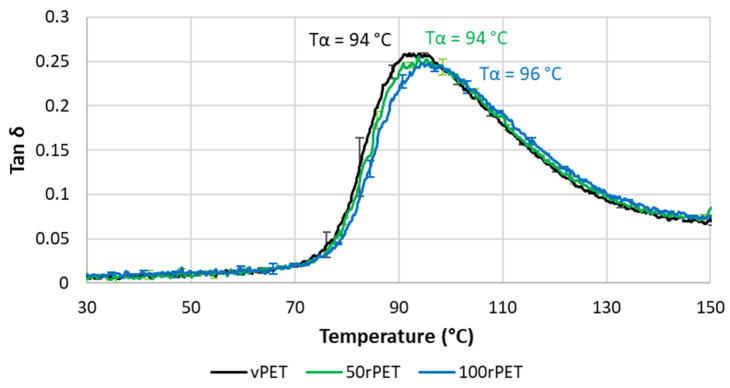
Phase angle (Tan δ) as a function of temperature for vPET, 50rPET, and 100rPET measured by DMA, in tensile mode, at a frequency of 1 Hz and a heating rate of 2 °C/min.

**Figure 14 polymers-15-04613-f014:**
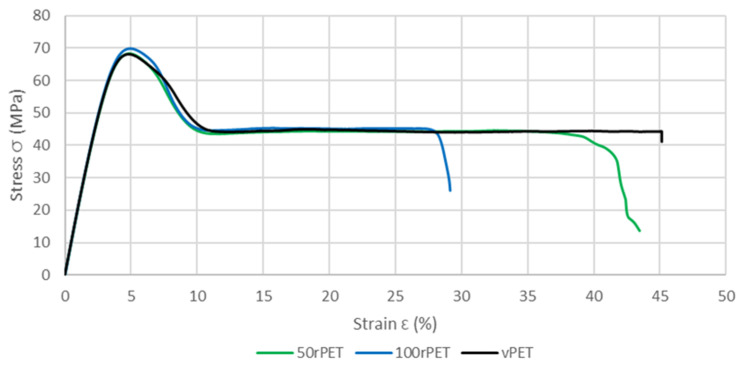
Stress–strain curves of standard samples for vPET, 50rPET, and 100rPET, obtained in uniaxial conditions until rupture, at room temperature and with a cross head velocity of 5 mm/min.

**Figure 15 polymers-15-04613-f015:**
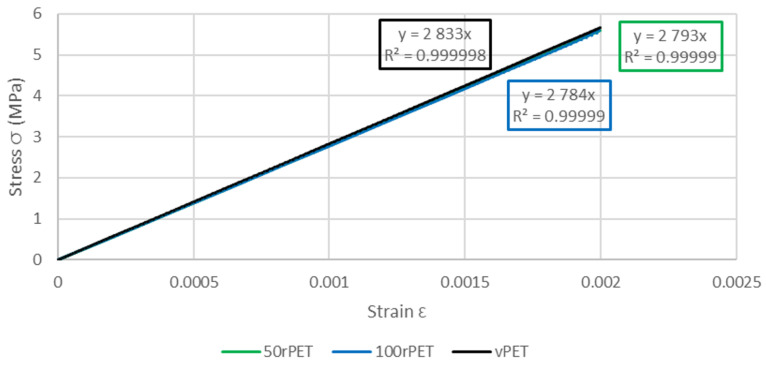
Zoom of stress–strain curves of standard samples for vPET, 50rPET, and 100rPET, until a deformation of 0.2%, obtained in uniaxial conditions with an extensometer to calculate Young’s modulus.

**Table 1 polymers-15-04613-t001:** Mn, Mw, and Polydispersity Index (PDI) for vPET and 100rPET at 35 °C.

Polymer	Mn (kg·mol^−1^)	Mw (kg·mol^−1^)	PDI
vPET	31.5 ± 0.1	53.8 ± 1.1	1.71 ± 0.04
100rPET	31.2 ± 1.6	44.9 ± 0.1	1.54 ± 0.08

**Table 2 polymers-15-04613-t002:** Melt Flow Index (MFI) for vPET and 100rPET at 270 °C.

Polymer	MFI (g/10 min)
vPET	20.7 ± 0.3
100rPET	36.5 ± 0.4

**Table 3 polymers-15-04613-t003:** AFM, DMA, and uniaxial tensile test modulus values for vPET, 50rPET, and 100rPET at room temperature.

Material	Max Modulus AFM (GPa)	Elastic Modulus (GPa)	Young’s Modulus (GPa)
vPET	2.13 ± 0.12	2.72 ± 0.09	2.83 ± 0.01
50rPET	3.03 ± 0.29	2.56 ± 0.06	2.79 ± 0.03
100rPET	2.13 ± 0.13	2.69 ± 0.03	2.78 ± 0.03

## Data Availability

Data are contained within the article.

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
