# Peer review of "A Comparative Study on Crystallisation for Virgin and Recycled Polyethylene Terephthalate (PET): Multiscale Effects on Physico-Mechanical Properties"

_polymers, 2023, doi:10.3390/polym15234613_

Round 1

Reviewer 1 Report (Previous Reviewer 3)

Comments and Suggestions for Authors

The manuscript reports A comparative study on the crystallization for virgin and recycled polyethylene terephtalate (PET): multi scale effects on phyisicomechanical properties, and there are several issues that need to be attended/corrected before continue the publication process, following they are detailed:

- Please correct the title of section 2.5, must be Differential Scanning Calorimetry (DSC), also in other parts where this term was used.

- which is the relevance of PDI? I mean this variable (reported in table 1)is not discussed in text.

- What is it means significantly lower the diminish of Mw (line274)? I mean how much has to be the difference for consider significantly? Also in case of MFI.

-Why for calculate the lamella thickness use the first heating step in DSC and not the second one? And for figure 5, what is it means population 1 and 2?

-which is the relationship between higher thickness and higher melting temperature? this according with results discussion commented in lines 392-393.

- how is the couplin with crystallineity ratios Xc measured by DSC and SAXS results?

- Please correct in Y axis must be Storage modulus no Elastic modulus figure 11.

- Line 506, how can relates the dispersion with DMA results? In other hand figure 12 was not cited in text.

- In lines 510-512 indicate that Tn delta peak shift to higher temperatures, but the difference is only 2ºC, which in thermal analysis can be consider negligible, please correct this.

- Table captions must be placed on the top of table.

-Please check instructions for authors in the case of references format.

Author Response

Reviewer 1

R1Q1. Please correct the title of section 2.5, must be Differential Scanning Calorimetry (DSC), also in other parts where this term was used.

R1A1.  This title has been modified (see line number 178 and the rest of the document checked.

R1Q2. Which is the relevance of PDI? I mean this variable (reported in table 1) is not discussed in text.

R1A2. The relevance of the polydispersity index (PDI) is presented in paragraph “2.3 Size Exclusion Chromatography (SEC)” on the line 155 As explained, this parameter “is defined as the ratio of Mw and Mn. If the value is close to 1, the molecular weight distribution is low”. Moreover, we precise that the PDI’s data are extracted from the table 1 when results are discussed (lines 276 and 282).

R1Q3. What is it means significantly lower the diminish of Mw (line 273-274)? I mean how much has to be the difference for consider significantly? Also in case of MFI.

R1A3. The difference is considered as significant when it exceeds the uncertainty of measurement.

R1Q4. Why for calculate the lamella thickness use the first heating step in DSC and not the second one? And for figure 6, what is it means population 0 and 2?

R1A4. Initially amorphous PETs have been crystallized in quiescent conditions, during an isothermal step of 140 °C (lines 138 to 144). These typical semi-crystalline microstructures have been used to perform all analysis (DSC, SAXS and AFM). As a result, the first heating step in DSC must then be considered. The second one would totally erase the thermo-mechanical paths and the associated microstructures developed. 

The two populations are defined at the lines 398-399.

R1Q5. Which is the relationship between higher thickness and higher melting temperature? this according with results discussion commented in lines 411-414.

R1A5. In fact, more thermal energy will be needed to melt thicker crystalline lamellaes.

The Gibbs-Thomson equation links melting temperature, Tm, to the lamellar thickness, lc. If Tm increases, the lamellar thickness increases as 1/(1-Tm/Tm0) increases (see modified text at lines 411-414).

R1Q6. How is the coupling with crystallinity ratios Xc measured by DSC and SAXS results?

R1A6. The crystallinity ratio Xc measured by DSC is used to calculate crystalline lamellas thickness (lines 432-435). A more rigorous approach would have been to use crystallinity ratio issued from X-ray scattering analyses. This point should be considered in a future study. As mentioned in the text on line 444, this approach just gives tendencies.

R1Q7. Please correct in Y axis must be Storage modulus no Elastic modulus figure 12.

R1A7. The legend has been modified (see lines 516, 520 and 529).

R1Q8. Line 531, how can relates the dispersion with DMA results? In other hand figure 13 was not cited in text.

R1A8. The trials were carried out 3 times for all materials. The error of measure obtained after these three trials is lower than the dispersion of temperature given by the DMA device. The figure 13 is cited at lines 529 and 544.

R1Q9. In lines 536-537 indicate that Tn delta peak shift to higher temperatures, but the difference is only 2ºC, which in thermal analysis can be consider negligible, please correct this.

R1A9. We consider that the precision of 1°C is low enough to justify the conclusion of our paragraph (see lines 545).

R1Q10. Table captions must be placed on the top of table.

R1A10. As recommended, table captions have been placed on the top of table.

R1Q11. Please check instructions for authors in the case of references format.

R1A11. We checked the references.

Reviewer 2 Report (New Reviewer)

Comments and Suggestions for Authors

The research was carried out very high quality and looks holistic. However, I would like the authors to supplement the study with IR spectroscopy data on the original and processed polymer with some explanations.

Undoubtedly, the article should be published

Author Response

Reviewer 2

R2Q1. The research was carried out very high quality and looks holistic. However, I would like the authors to supplement the study with IR spectroscopy data on the original and processed polymer with some explanations.

R2A1.  In our study, we observed the difference between materials (virgin and recycled PET) and not the differences observed after processing. However, it will be very interesting work and we can think of this in future work.

Undoubtedly, the article should be published.

Thank you for your encouragement.

Reviewer 3 Report (New Reviewer)

Comments and Suggestions for Authors

The manuscript entitled “A comparative study on crystallisation for virgin and recycled polyethylene terephthalate (PET): Multiscale effects on physico-mechanical properties” reports the studies between virgin and recycled PET. The manuscript is well-prepared, interesting, and all data obtained from the measurements are well described and analyzed. I think the manuscript can be published in the Journal after very minor amendments:

1.            Page 3, line 135: The glass transition temperature should be added.

2.            Page 4, line 160: Was the HFIP bought? Add information about the seller and the purity.

3.            Is the first heating shown in Fig. 1 (and the results in Fig. 3) the heating of a “virgin” sample that has not previously been melted? If yes, then some differences between the very first heating and the next ones should be expected to appear.

4.            Page 6, lines 264-266: The sentences given there should be (and have to be) deleted. (I think that they are some sentences from the template).

5.            Tables 1-2 (p.7), 4 (p.17); Figs. 6 (p.11), 8 (p.13), 9-10 (p.14), 13-14 (p.17): The temperatures should be added (for Table 2 – 270 deg. C, for Fig. 6 – at room temperature).

6.            Figures 13, 14 (p.9): (i) The author should add (in the figure captions): The thermograms are vertically shifted. (ii) On the vertical axes there are numerical values with the comma between the integer and decimal parts (there should be the point); (iii) The authors should describe the complex two-peak anomalies for 50rPET and 100rPET (and vPET – the two-peak anomaly is not well resolved, I think so). (iv) The values of the enthalpy change should be calculated and added for all anomalies observed. (v) There is a lack of thermograms for the cooling process. Are any anomalies observed (the crystallization, vitrification process, etc.)? This information is crucial.

7.            Figure 6: Add the errors for the estimated periodicity values.

8.            Figures 8-10: I think that the parameters of the Gaussian shown in Figures 8-10(d) can be added (in a table) to the manuscript. The mean values are given (in the text on pages 14-15), but the standard deviations or R-square parameter can be very helpful for a potential reader.

9.            Minor errors found (mainly editorial type): (i) page 2, line 61: there is “(…) as imposed by Europe [21].”, and there should be “(…) as imposed by the European Union [21].” (or “the European Union Parliament”); (ii) page 3, lines 111 and 117: I think that the meanings of 100rPET and 50rPET can be given, i.e. 100% recycled PET or 50% recycled PET; (iii) Figure 12, the vertical axis: there are commas instead of points (between the integer and decimal parts of the values).

Author Response

Reviewer 3

The manuscript entitled “A comparative study on crystallisation for virgin and recycled polyethylene terephthalate (PET): Multiscale effects on physico-mechanical properties” reports the studies between virgin and recycled PET. The manuscript is well-prepared, interesting, and all data obtained from the measurements are well described and analyzed. I think the manuscript can be published in the Journal after very minor amendments:

R3Q1. Page 3, line 133: The glass transition temperature should be added.

R3A1.  We added a bibliographic reference on the range of the PET glass transition temperature (see line 133).

R3Q2. Page 4, line 158: Was the HFIP bought? Add information about the seller and the purity.

R3A2.  The SEC experiments were outsourced at the INSA Valor company as mentioned on lines 163-164. We don’t have any information about the origin of HFIP and about its purity.

R3Q3. Is the first heating shown in Fig. 1 (and the results in Fig. 3) the heating of a “virgin” sample that has not previously been melted? If yes, then some differences between the very first heating and the next ones should be expected to appear.

R3A3. Yes, the first heating step is presented in Figure 1.

In Figure 3, during the first heating, we analyze the crystallized preforms that have been submitted to a thermal treatment at 140°C, to analyze the induced crystallinity. The first heating step leads to a melting of the material. A cooling step is then imposed from the melt with the same sample at a cooling rate of 10°C/min. A last analysis during a second heating step follows the cooling step. As thermal paths are different between the first and second heating steps, we expect differences that are commented on lines 362 to 375.

R3Q4. Page 6, lines 265-266: The sentences given there should be (and have to be) deleted. (I think that they are some sentences from the template).

R3A4. The sentences have been removed (lines 265 to 266).

R3Q5. Tables 1-2 (p.7), 4 (p.18); Figs. 7 (p.13), 9-10 (p.15), 11 (p.16), 15 (p.18), 15 (p.19): The temperatures should be added (for Table 2 – 270 deg. C, for Fig. 7 – at room temperature).

R3A5. As recommended, all this information has been added on tables and figures legend.

R3Q6. Figures 3 (p. 9), 4 (p.10): (i) The author should add (in the figure captions): The thermograms are vertically shifted. (ii) On the vertical axes there are numerical values with the comma between the integer and decimal parts (there should be the point); (iii) The authors should describe the complex two-peak anomalies for 50rPET and 100rPET (and vPET – the two-peak anomaly is not well resolved, I think so). (iv) The values of the enthalpy change should be calculated and added for all anomalies observed. (v) There is a lack of thermograms for the cooling process. Are any anomalies observed (the crystallization, vitrification process, etc.)? This information is crucial.

R3A6. (i) As recommended, we added the sentence: “The thermograms are vertically shifted”.

R3Q7. Figure 7: Add the errors for the estimated periodicity values.

R3A7. The errors were added to figure 7.

R3Q8. Figures 9-11: I think that the parameters of the Gaussian shown in Figures 9-11(d) can be added (in a table) to the manuscript. The mean values are given (in the text on pages 16), but the standard deviations or R-square parameter can be very helpful for a potential reader.

R3A8. Yes, these parameters are important. Average and standard deviations are shown in Table 4. For the R-squared parameter, a sentence has been added in the text: “The Gaussian fit was successfully applied to each histogram with an R-squared val-ue greater than 0.9” (lines 477-478).

R3Q9. Minor errors found (mainly editorial type): (i) page 2, line 61: there is “(…) as imposed by Europe [21].”, and there should be “(…) as imposed by the European Union [21].” (or “the European Union Parliament”); (ii) page 3, lines 112 and 115: I think that the meanings of 100rPET and 50rPET can be given, i.e., 100% recycled PET or 50% recycled PET; (iii) Figure 13, the vertical axis: there are commas instead of points (between the integer and decimal parts of the values).

R3A9. All the recommendations were considered.

Round 2

Reviewer 1 Report (Previous Reviewer 3)

Comments and Suggestions for Authors

After review the corrected version, still there are some details that need to be corrected to continue the publication process, following they are detailed:

About R1A4:I think it is a confusion about the first and second heating cycle in DSC, usually the first cycle is used to delete the thermal history of materials, so please check this.

R1A8: this is not clear enough, the 3 times measurements for DMA are related to reproducibility, my question is about how the dispersion can be determinated using the obtained results by DMA. i means usually some signals (Tan delta or Storage modulus) can be used to discussion about the dispersion of fillers, reinforcers or other kind of additives in polymer matrix.

R1A9: About the difference of 2ºC in Tan delta peak, it is not an authors consideration, the theory support this, for consider that a property has a considerable variation, this must be at least 4-5ºC.

Author Response

Reviewer 1

R1Q4. Why for calculate the lamella thickness use the first heating step in DSC and not the second one? And for figure 6, what is it means population 0 and 2?

R1A4. Initially amorphous PETs have been crystallized in quiescent conditions, during an isothermal step of 140 °C (lines 138 to 144). These typical semi-crystalline microstructures have been used to perform all analysis (DSC, SAXS and AFM). As a result, the first heating step in DSC must then be considered. The second one would totally erase the thermo-mechanical paths and the associated microstructures developed. 

The two populations are defined at the lines 398-399.

R1Q4.2: I think it is a confusion about the first and second heating cycle in DSC, usually the first cycle is used to delete the thermal history of materials, so please check this.

R1A4.2: Yes, usually, to study the properties of materials, the scientists examine the second melting process to erase the thermal history. This information interest us and we examine this part to see the behavior of our material under the same classical cooling speed (-10°C/min).

However, in this study, we want to compare the crystallization at 140°C of our materials prepared in the same condition. These preforms were used for the mechanical characterizations and the AFM measurement. Using the second melting process by DSC does not give us the information about the crystallinity of these same preforms used in the other experiments. 

We would be able to undergo our samples at same conditions of preparation of our crystalline preforms but it is the same thing that taking directly a part of our crystalline preforms.

R1Q8. Line 531, how can relates the dispersion with DMA results? In other hand figure 13 was not cited in text.

R1A8. The trials were carried out 3 times for all materials. The error of measure obtained after these three trials is lower than the dispersion of temperature given by the DMA device. The figure 13 is cited at lines 529 and 544.

R1Q8.2: this is not clear enough, the 3 times measurements for DMA are related to reproducibility, my question is about how the dispersion can be determinated using the obtained results by DMA. i means usually some signals (Tan delta or Storage modulus) can be used to discussion about the dispersion of fillers, reinforcers or other kind of additives in polymer matrix.

R1A8.2: In this study, we don’t add any fillers. Moreover, we use commercial grade and we have any idea about their exact compositions. We can’t link the dispersion of DMA with fillers or contaminants.

R1Q9. In lines 536-537 indicate that Tn delta peak shift to higher temperatures, but the difference is only 2ºC, which in thermal analysis can be consider negligible, please correct this.

R1A9. We consider that the precision of 1°C is low enough to justify the conclusion of our paragraph (see lines 545).

R1Q9.2: About the difference of 2ºC in Tan delta peak, it is not an authors consideration, the theory support this, for consider that a property has a considerable variation, this must be at least 4-5ºC.

R1A9.2: After this second revision and agreement in all the authors, we consider that effectively the differences of 2°C is not significant. We just conserve the passage about the area under the curve.

This manuscript is a resubmission of an earlier submission. The following is a list of the peer review reports and author responses from that submission.

Round 1

Reviewer 1 Report

Comments and Suggestions for Authors

1. The introduction can be supplemented by reviews concerning the methods of recycling and upcycling of PET (10.3390/membranes12111105), obtaining composites based on rPET (10.1016/j.compositesb.2021.108928).

2. Calorimetric data and physical-mechanical properties for rPET, vPET and their mixtures are given in Velásquez (10.1016/j.clay.2019.105185). From my point of view, your data should at least be compared with the data given in this study. Also of interest to the authors may be the work of Celik (10.3390/polym14071326).

3. The conclusion should be shortened and specified.

Author Response

R1Q1. The introduction can be supplemented by reviews concerning the methods of recycling and upcycling of PET (10.3390/membranes12111105), obtaining composites based on rPET (10.1016/j.compositesb.2021.108928).

R1A1.  The introduction has been modified, including the recommendations with both bibliographic references (see lines number 49 to 50 and 57).

R1Q2.Calorimetric data and physical-mechanical properties for rPET, vPET and their mixtures are given in Velásquez (10.1016/j.clay.2019.105185). From my point of view, your data should at least be compared with the data given in this study. Also of interest to the authors may be the work of Celik (10.3390/polym14071326).

R1A2. We have added your recommendations about the work of Celik (see lines number 338 and 341). Concerning Velásquez’ results, the thermal and mechanical properties analysis deals with amorphous PET films that were extruded. In our study, all characterizations are performed on semi-crystalline PET, especially on massive and injected parts. As a result, comparing the microstructural organization and the melting temperatures associated to amorphous and semi-crystalline polymer, and also to extruded and injected material appears dangerous.

R1Q3. The conclusion should be shortened and specified.

R1A3. As recommended, the conclusion has been shortened and specified (see lines number 553 to 562).

Reviewer 2 Report

Comments and Suggestions for Authors

The article "A comparative study on crystallization for virgin and recycled polyethylene terephthalate (PET): Multiscale effects on physicomechanical properties" seems to be an interesting topic. However, a simple, quiet reading makes many inconsistencies and flaws emerge that recommend rejecting the article. To notice that are plenty of additional minor questions to solve. For instance, the lack of information about the preforms' dimensions, the way they obtain the champions for measurements maintaining a rectangular shape, and so on.

But many others respond to an absolute lack of knowledge about what a polymer blend means. That is, the mixing obtained by the screw of the injection machine cannot reach an intimate mixture between two different samples (vPET and 100rPET) to obtain a homogeneous mixture of them (50rPET). NOTE THAT THIS IS A FUNDAMENTAL ISSUE IN BASIC POLYMER PROCESSING.

However, being this fact enough to reject the article, there exists an even more important flaw as it is the use of the crystalline content obtained by DSC to calculate the lamellae thickness using L values obtained by SAXS. Please, be serious, and DO NOT FALL INTO THE TRAP OF COMPARING APPLES AND ORANGES. The authors should know that the determination of the crystalline content is different when calculated from thermal analysis than when obtained by WAXS, and so, there is non-sense to comparing Xray values (SAXS) with thermal analysis (DSC) buy to other Xray based values as the crystalline content obtained by WAXS. This reviewer wonders why the authors have not explored this possibility rather than comparing incompatible data.

There are many other concerns in the text, but the ones mentioned earlier are so important as to reject the paper for being fake.

Author Response

R2Q1. The article "A comparative study on crystallization for virgin and recycled polyethylene terephthalate (PET): Multiscale effects on physico-mechanical properties" seems to be an interesting topic. However, a simple, quiet reading makes many inconsistencies and flaws emerge that recommend rejecting the article. To notice that are plenty of additional minor questions to solve. For instance, the lack of information about the preforms' dimensions, the way they obtain the champions for measurements maintaining a rectangular shape, and so on.

R2A1. The dimensions of the preform geometry have been added in the text (see lines number 115 to 117). Sidel Group has not authorized us to publish more details about the preform design. Rough information are then given here which enables to understand the basic form of the injected part.

For the collection of the DMA samples, the core of preform has been machined, removing the around parts. A second step which consists in polishing the two surfaces (inside and outside) has been added. The way the sample has been collected is now described in the text (see lines number 234 to 237).

R2Q2. But many others respond to an absolute lack of knowledge about what a polymer blend means. That is, the mixing obtained by the screw of the injection machine cannot reach an intimate mixture between two different samples (vPET and 100rPET) to obtain a homogeneous mixture of them (50rPET). NOTE THAT THIS IS A FUNDAMENTAL ISSUE IN BASIC POLYMER PROCESSING.

R2A2. The injection process has been used in a purely industrial context, involving a mixing inside the screw of the injection machine commonly used at the industrial scale when recycled polymers are added to virgin one. It was clearly a choice that we made to use representative preforms that can be found today in the food packaging market.

Moreover, no inhomogeneity trace is obvious on injected parts in the visible radiation range. The blue colour remains homogeneous in the whole injected parts. These observations are confirmed thanks to optical microscopy and Scanning Electron Microscope (SEM) where no presence of different phases could be observed at all. Besides, at the macroscopic scale, no prematurely failure took place in the case of the PET blend. The mechanical signature described with AFM analysis does not present any division between the two materials presence too. The total miscibility, that should be expected for PET blends, is obvious. The goal of the publication is exactly to show that larger distribution of mechanical properties is present with the blend which involves probably different homopolymers or random copolymers. Nevertheless, no clear discrepancies in the local microstructural areas are observed testifying of the physico-mechanical properties homogeneity of the blend (see lines number 485 to 487).  

R2Q3. However, being this fact enough to reject the article, there exists an even more important flaw as it is the use of the crystalline content obtained by DSC to calculate the lamellae thickness using L values obtained by SAXS. Please, be serious, and DO NOT FALL INTO THE TRAP OF COMPARING APPLES AND ORANGES. The authors should know that the determination of the crystalline content is different when calculated from thermal analysis than when obtained by WAXS, and so, there is non-sense to comparing X-ray values (SAXS) with thermal analysis (DSC) buy to other X-ray based values as the crystalline content obtained by WAXS. This reviewer wonders why the authors have not explored this possibility rather than comparing incompatible data.

R2A3. The aim of this approach is not to compare absolute values obtained with two different methods but rather to check, or not, if the same trends can be observed with the two methods. Rigorously, and for a given method analysis, the same protocols have been used for the three materials. Therefore, the SAXS coupled with DSC values give an estimation, an order of magnitude and a trend but not an absolute result. In a future work, it is true that it would be more confident to calculate the crystallinity ratio thanks to WAXS analysis. We have re-organized the conclusions description on this paragraph to avoid any confusion and only describe the trends obtained (see lines number 411 to 420).

R2Q4. There are many other concerns in the text, but the ones mentioned earlier are so important as to reject the paper for being fake.

R2A4. We hope that the previous answers have managed to convince you of the relevance of the publication.

Reviewer 3 Report

Comments and Suggestions for Authors

The manuscript reports the comparative study on crystallization for virgin and recycled PE: multi scale effects on physics-mechanical properties, it is a well structured work but there are some specific comments that need to be corrected before continue the publication process, following they are detailed:

-First time an abbreviation is written must be defined, for instance; PP, PE, PVC, etc. And one, the abbreviation was defined must be used in whole the document.

-please indicate the geometry preform obtained after injection.

-For Tcc and Tm was it used the second heating cycle or the first?

Is it the uni axial tensile test a DMA ? I mean the concepto of DMA technique is the often an oscillating stress or strain is applied but in this case the load is applied in normal mode.

-in figure 3 and 4, please delete decimals after point for all temperatures reported (Tg, Tm), due in thermal analysis they are consider negligible when difference is lees than 5ºC.

-In some of the results discussion indicate that contaminants presents are responsable for the crystallization process, but what kind of contaminants are those?

-in figure 12, please delete decimal after point for temperatures reported, due in thermal analysis they can consider negligible.

discussion in 506 507 is not well supported, due in thermal analysis a difference of 2 ºC is considered negligible

-For references please follow the instructions for authors.

Author Response

The manuscript reports the comparative study on crystallization for virgin and recycled PE: multi scale effects on physics-mechanical properties, it is a well-structured work but there are some specific comments that need to be corrected before continue the publication process, following they are detailed:

R3Q1. First time an abbreviation is written must be defined, for instance; PP, PE, PVC, etc. And one, the abbreviation was defined must be used in whole the document.

R3A1. The text has been modified (see lines number 67), including the recommendations with the definition of the abbreviations.

R3Q2. please indicate the geometry preform obtained after injection.

R3A2. The dimensions of the preform geometry have been added in the text (see lines number 115 to 117). Sidel Group has not authorized us to publish more details about the preform design. Rough information are then given here which enables to understand the basic form of the injected part.

R3Q3. For Tcc and Tm was it used the second heating cycle or the first?

R3A3. For the figure 3, it is the first heating step. For figure 4, it is the second one. The legends associated to the figures 3 and 4 are detailing these points. The used Tm are the melting temperatures issuing from the first heating ramp.

R3Q4. Is it the uniaxial tensile test a DMA ? I mean the concept of DMA technique is the often an oscillating stress or strain is applied but in this case the load is applied in normal mode.

R3A4. Yes, the uniaxial tensile test has anything to do with DMA, it is an error. The title of the paragraph “Uni-axial tensile test” has been modified (see lines number 252).

R3Q5. in figure 3 and 4, please delete decimals after point for all temperatures reported (Tg, Tm), due in thermal analysis they are consider negligible when difference is less than 5ºC.

R3A5. The decimals after point have been removed on the two figures, 3 and 4.

R3Q6. In some of the results discussion indicate that contaminants presents are responsible for the crystallization process, but what kind of contaminants are those?

R3A6. Great question! Currently, a phD project aims at identifying the nature of the contaminants; it is an on-going work. No polyolefins are present obviously and most of the identified contaminants should mainly be due to thermo- and mechanical degradation of PET. We have not added these results to the present publication as they might be published in another context and elsewhere.

R3Q7. in figure 12, please delete decimal after point for temperatures reported, due in thermal analysis they can consider negligible.

R3A7. The decimals after point have been removed on the figure 12.

R3Q8. discussion in 506 507 is not well supported, due in thermal analysis a difference of 2 ºC is considered negligible

R3A8. Indeed, the difference is very low but it is not really negligeable. The uncertainty on the measurement of the maximum of Tan delta peak is of 1 °C. So that we can consider that rPET does present a slightly higher alpha transition temperature, which is also confirmed by DSC analysis. This result is also in accordance with previous work (Rohart et al. [42]). No difference is visible between the blend and the virgin PET.

R3Q9. For references please follow the instructions for authors.

R3A9. the references have been made in the correct format (ACS).

Round 2

Reviewer 2 Report

Comments and Suggestions for Authors
